# Monitoring Psychometric States of Recovery to Improve Performance in Soccer Players: A Brief Review

**DOI:** 10.3390/ijerph19159385

**Published:** 2022-07-31

**Authors:** Okba Selmi, Ibrahim Ouergui, Antonella Muscella, Giulia My, Santo Marsigliante, Hadi Nobari, Katsuhiko Suzuki, Anissa Bouassida

**Affiliations:** 1High Institute of Sports and Physical Education of Kef, University of Jendouba, Jendouba 7100, Tunisia; ouergui.brahim@yahoo.fr (I.O.); bouassida_anissa@yahoo.fr (A.B.); 2High Institute of Sports and Physical Education, Ksar Said, University of Manouba, Tunis 2010, Tunisia; 3Department of Biological and Environmental Science and Technologies, University of Salento, 73100 Lecce, Italy; antonella.muscella@unisalento.it (A.M.); giulia.my@unisalento.it (G.M.); santo.marsigliante@unisalento.it (S.M.); 4Faculty of Physiology, School of Sport Sciences, University of Extremadura, 10003 Cáceres, Spain; hadi.nobari1@gmail.com or; 5Department of Exercise Physiology, Faculty of Educational Sciences and Psychology, University of Mohaghegh Ardabili, Ardabil 56199-11367, Iran; 6Department of Motor Performance, Faculty of Physical Education and Mountain Sports, Transilvania University of Braşov, 500068 Braşov, Romania; 7Faculty of Sport Sciences, Institute of Sports Nutrition, Waseda University, Tokyo 359-1192, Japan

**Keywords:** soccer, well-being, training load, fatigue, performance, TQR

## Abstract

In order to maximize adaptations and to avoid nonfunctional overreaching syndrome or noncontact injury, coaches in high-performance sports must regularly monitor recovery before and after competitions/important training sessions and maintain well-being status. Therefore, quantifying and evaluating psychometric states of recovery during the season in sports teams such as soccer is important. Over the last years, there has been substantial growth in research related to psychometric states of recovery in soccer. The increase in research on this topic is coincident with the increase in popularity obtained by subjective monitoring of the pre-fatigue state of the players before each training sessions or match with a strong emphasis on the effects of well-being or recovery state. Among the subjective methods for players’ control, the Hooper index (HI) assesses the quality of sleep during the previous night, overall stress, fatigue, and delayed-onset muscle soreness. Additionally, the total quality of recovery (TQR) scale measures recovery status. The HI and TQR recorded before each training session or match were affected by the variability of training load (TL) and influenced the physical and technical performances, and the affective aspects of soccer players. Researchers have recommended wellness monitoring soccer players’ psychometric state of recovery before each training session or match in order to detect early signs of fatigue and optimize high-level training performance. This method allows for better detecting signs of individual fatigue and allows coaches to adapt and readjust the TL, and avoid physical and technical gaps in order to improve the performance of soccer players.

## 1. Introduction

Monitoring psychometric states of stress, fatigue, and recovery in athletes is becoming increasingly critical in high-level soccer in order to maximize recovery and support peak performance [1,2]. Among other things, even in a return-to-play (RTP) scenario, it has become fundamental for elite athletes after an acute trauma, given the economic and competitive interweaving associated with the downtime of professional players, to recognize the psychological facet of RTP after an injury [3]. Several psychometric indices have assessed the relationship between the subjective states of well-being, recovery, and mood with physical, technical, tactical, and psychological outcomes in elite soccer athletes [1,4,5,6,7,8,9,10,11,12,13,14]. Indeed, increasing training load (TL), an accumulation of fatigue, an imbalance between the training–recovery process, and player fitness can all influence a player’s physical, technical, tactical, and psychological reactions to a training stimulus [11,15,16,17,18]. In this regard, increased pre-fatigue states and poor recovery during training can reduce performance and induce negative psychological outcomes in participants during training sessions and competitions [8,11,19,20,21,22].

For this reason, researchers have recommended monitoring soccer players’ psychometric state of recovery to detect early signs of fatigue and optimize high-level training performance [3,4,9,14,19]. For example, the Hooper index (HI) assesses quality of sleep during the previous night, overall stress, fatigue, and delayed-onset muscle soreness (DOMS) using four items on a 7-point Likert scale [21]. Additionally, the total quality of recovery (TQR) scale is a validated psychometric tool used to measure recovery status [23]. These measures are used to express the subjective state of a player or team during training or competition [15,16,24]. HI subscores (i.e., sleep quality, overall stress, fatigue, and DOMS) and TQR scores are significant predictors of performance in soccer athletes [4,5,6,11,22,25], with increasing HI and TQR scores significantly associated with heavy training loads [8,11,14,15,16,21,25,26]. Correspondingly, lack of recovery, poor quality of sleep from the previous night, and increased levels of stress, fatigue, and DOMS [4,21,23,25] during training can negatively impact athletic performance [11,16,21]. Thus, monitoring psychometric states of recovery in soccer players via the HI and TQR may allow coaches and trainers to properly program and adapt training loads in order to maximize performance and reduce the risk of injury, overtraining, and nonfunctional overreaching [21,22,23,24,25,26,27].

This brief review summarizes the current literature on the relationship between soccer players’ psychometric state of recovery with the following outcomes: (1) TL; (2) technical, physiological, and physical performance; and (3) ratings of perceived exertion (RPE) and enjoyment. Additionally, a discussion on the impact of Ramadan on subjective metrics of stress, fatigue, and recovery is presented. Practical applications and future research are suggested.

## 2. Materials and Methods

### 2.1. Search Strategy

This review incorporated studies that examined the monitoring of psychometric status of soccer players to detect early signs of fatigue and to optimize high-level training performance. A literature search was performed independently by the authors using PubMed, Scopus, WoS, and Google Scholar databases from inception up to 1 April 2022. Moreover, we performed manual searches of relevant journals and reference lists obtained from published articles. Electronic databases were searched using the keywords “football” or “soccer” and “psychometric status,” “well-being indices,” “total quality of recovery,” “recovery” or “Hooper index”; “training load” or “fatigue”. We used indexing words and free search terms, which we clustered according to the PICo scheme (population, phenomenon of interest, context) [28]. The inclusion criteria for these articles were: (1) data concerning physical and psychological evaluation, statistical compilation, or time–motion analysis; (2) players included professional male soccer players; (3) studies examined HI (sleep, stress, fatigue, and DOMS) or TQR; and (4) the original studies were published in English. Studies were excluded if they: (1) included sports other than soccer or (2) did not consist of published, primary research data (i.e., reviews, commentaries, interviews or expert opinions, posters, or book chapters were not included).

Selected authors (Okba Selmi, Giulia My, Antonella Muscella) independently extracted and reviewed study data to determine if a given study met the inclusion criteria. Disagreements about whether the inclusion criteria were met were resolved by discussion with the other authors.

### 2.2. The Hooper Index (HI)

The literature study showed that the HI, when included, was used in individual sports [21,29] and in team sports, particularly in soccer [4,5,6,10,11,15,20,25,30,31]. HI is measured using rating scales on sleep quality, stress, level of fatigue, and DOMS [21]. HI is a tool based on self-assessment scales relating to the current state of players to determine psycho-physiological scores corresponding to sleep, fatigue, stress, and DOMS [5,6,11,15,20,21]. Each of these variables was measured in the morning or 30–15 min before each first daily training session using scales of 1–7. For sleep quality, the scale ranges from 1 (very, very good) to 7 (very, very bad) and then for the other qualities (fatigue, stress, and muscle pain), the scale starts at 1 (very, very low) and ranges to 7 (very, very high) (Table 1). The HI can be calculated by summing the scores of these 4 subjective subscales [21] (Table 1).

Recent research has reported that the HI is considered as one of the most interesting markers to obtain information on the psycho-physiological state of soccer players [4,9,10,11]. For example, Selmi et al. [16] indicated that this method is used in soccer to examine the current state of players and to control the emotional changes in relation to players’ TL. Similarly, other studies have considered this tool as a very important index for the detection of decreased performance, disturbance of physiological state, immune dysfunction, and hormonal imbalance [20,32].

This method allows better detection of signs of individual fatigue when interpreted with training duration and intensity [15,16,21]. It also allows coaches and physical coaches to better program and choose the types and qualities of exercises in players and to plan the TL with precision in order to achieve better performance [4,11,15,21,25,31,32,33,34].

### 2.3. Total Quality of Recovery (TQR)

Bishop et al. [35] identified recovery as the vital period between successive training sessions or competitions with the potential to improve performance later and highlighted the fact that players will spend more time recovering what they achieve in training. This concept is defined as the psycho-physiological process for reducing fatigue and regaining vigor [16]. Owing to the lack of available instruments, it was necessary to develop a new method to measure psycho-physiological recovery in order to find the balance between training and recovery [23,36,37,38,39]. The TQR method was therefore tightly organized around the concept of RPE in order to accentuate the relationship between training and recovery [23].

The main purpose of performing TQR was to prevent the onset of fatigue and seek a balance between training and recovery [38]. This tool is an overall measure of perception of recovery on a 6–20 point scale, estimating the perception of recovery daily to detect the current form of athletes and to monitor short-lived emotional changes [25]. The TQR scale (Table 2) is similar to the perceived exertion scale [40]. This tool is a means used to measure psycho-physiological recovery [36,41].

To obtain the perception of recovery score, players are called up before training to give an overall psycho-physiological score (physically and mentally) for the past 24 h, including the previous night’s sleep. This supports the subjective perception of player recovery in a qualitative way. Thus, it would have to be practiced basically to detect intra-individual changes [23].

A score for the TQR is ideal at a value of 20 [23] (Table 2) and a score of 13 is considered the minimum score; any scores below this indicate that recovery is incomplete [25]. Always keep in mind that scores vary from player to player over a period [7]. The measurement of recovery on a scale of 6 to 20 points gives to the technical staff and the players a collection of data and a simple and effective control of the level of player recovery without carrying out more complex or cumbersome tests [41]. It is so simple that recovery assessment can be done daily and easily estimated by the coach [42] (Table 2).

The additional value of TQR could also be used not only for performance prediction, but also for the prevention of overtraining and injuries [36,43]. The TQR score was used successfully in the prediction of athlete performance [41]. The use of TQR is not only used as a tool to detect overtraining syndromes, but also to attract the attention of the players and the coaches to the importance and relevance of recovery after periods and training sessions. Consequently, the TQR is therefore a method which makes it possible to reduce the risks for the athlete of reaching overtraining [23].

The process of using the TQR scale should focus on player concentration and on the value of recovery rather than training and performance evaluation [41]. This tool can assist in preventing the onset of fatigue syndromes, although this subjective assessment could simply be influenced by external factors such as the place, the time, and the way of evaluating the athletes.

## 3. Results

### 3.1. Relationship between HI, TQR, and Internal Training Load

This method has been widely studied in soccer in different situations (age, level, sex, training period, workout, and training charge) [5,6,8,11,12,15,16,20,21,25,30].

The TQR and HI have often been used specifically to determine the quality of recovery and the current form of players in different disciplines, particularly in soccer [10,12,30,44,45].

TQR and HI are tools used to monitor the psycho-physiological state of players during training [11,41,43,46]. For this reason, most scientific research supports the idea that the follow-up of the process of recovery and the well-being of the players contribute to improving the ability to monitor the training charge and performance of players [12,32]. In this context, Selmi et al. 2018 [47] reported that TQR and HI are considered sensitive markers to monitor the effects of TLs and sensitive to stress identification and recovery [15].

Indeed, Selmi et al. [3] recommended using HI and TQR to investigate well-being as a common practice for detecting signs of pre-fatigue and monitoring physiological and psychometric status in soccer players. These devices are sensitive to the effects of TL [11]. For this reason, several scientific research projects have studied the relationship between these markers and TL. For example, Moalla et al. [15] studied the relationship between daily TL and the HI, reflecting their perceived quality of sleep, fatigue, stress, and DOMS during a 16-week training period in professional soccer players. They showed that significant relationships (*p* < 0.01) between TL and perceived sleep stress, fatigue, and DOMS (all *p* < 0.01). This finding indicated that TL affects well-being status of professional soccer players. The authors suggested that the HI, which is measured before daily training sessions, allows the coaches to better detect individual signs of pre-fatigue and therefore eventually accurately adapt the scheduled TL of the day considering the players’ status to reach optimal performance and avoid overtraining. Similarly, Selmi et al. 2018 [16] examined the effect of a 4 week intense training cycle during pre-season on TL variables and psychometric status (i.e., sleep, stress, fatigue DOMS, and TQR) and to examine relationships between these parameters. The major findings of this study showed that TL, monotony, and strain increased progressively during the 4 week intense training cycle. These variables in TL were related to TQR and HI. This study was conducted to support the interest of stress, fatigue, sleep, DOMS, and TQR measured before the daily first training session and TL as a simple noninvasive, nonfatiguing, sensitive, and effective for helping coaching staff to control soccer players’ psychometric status during intense training period. In the same perspective, Lathlean et al. [48] studied the associations between TL and HI in elite junior Australian soccer players across one competitive season. It was found that, over the season, TL had a significant association with stress and DOMS. The authors suggested that TL is important in managing the well-being of young soccer players. They indicated that quantifying loads and wellness at this level will help optimize player management and has the potential to reduce the risk of adverse events such as injury. Moreover, Clemente et al. [9] assessed positional playing differences on TL, session-RPE, and wellness across two different training microcycles (1 vs. 2 competitive games) and studied the association between TL and HI across an entire season among professional soccer players. Results showed significant correlations between TL and HI across two-game weeks indicating significantly higher score of fatigue and DOMS in weeks with two official matches. The authors suggest that TL may be sensitive to HI and vice versa.

Recently, the study of Nobari et al. [9] monitored the acute and chronic TL, monotony, and strain over a season and its relationships with HI in young soccer players. The result showed that values of acute, chronic load, and training strain were higher in the mid-competitive season and were lower in the start-competitive season. Furthermore, values of weekly fatigue, stress level, and DOMS were higher in the end-competitive season, and the sleep quality and stress were lower in the start-season while the fatigue and DOMS score were lower in the mid-season. The authors showed also that acute load, monotony, and strain related to sleep, stress, fatigue, and DOMS, indicating that HI was the best predictor of the acute load.

The goal of Malone et al.’s research [49] was to observe the impact of player well-being on the training output of elite footballers. Forty-eight footballers (age: 25.3 ± 3.1 years; height: 183 ± 7 cm; mass: 72 ± 7 kg) were involved in this observational study.

The results revealed a significant reduction in the well-being score, which had an impact on external and integrated training load ratios. In addition, during the physical tests, total distance reduction at high speed, distance at high speed, and maximum speed were measured.

Although female soccer players are on the rise, the main evidence on training monitoring and well-being in soccer has been related to men.

Owing to natural biological differences, more research is needed to understand the mechanisms of how they approach the training process. Therefore, descriptive studies would be valuable to characterize the reality of the training process in women’s football. In this context, Fermandes et al. [26,44] described the weekly variations of training monotony and chronic workload ratio through session-rated perceived exertion (s-RPE)) and compared s-RPE and variations of the Hooper index (stress, fatigue, DOMS, and sleep), with training and match days, from the same women’s Portuguese League team. The results showed that there were some associations between the Hooper index categories and s-RPE, such as stress or fatigue, stress or DOMS, stress or s-RPE, and fatigue or DOMS. Moreover, any in-season variations concerning internal load and perceived wellness seems independent of position or status in players [26]. The data also showed that the higher the players’ reported stress, the lower the observed s-RPE, thus possibly indicating a mutual interference of experienced stress levels on the assimilation of training intensity by elite women soccer players [26].

When s-RPE and HI data were analyzed in relation to the number of days away from the competitive one-match week (i.e., match day minus, MD-) with three training sessions a week (MD-5, MD-4, MD-2), the results showed that there is a higher TL on the first and second training sessions of the week (MD-5 and MD-4) than in the last training session (MD-2). Furthermore, matches presented the highest internal workload [44]. The results from these studies should not be generalized because the small sample size does not allow generalization of the results; in addition, it has been reported that in female players, a higher workload is performed by players at higher competition level compared with players at moderate levels [26,44].

Overall, these results suggest that the HI and TQR can be useful in planning training loads during periods of training. A summary of studies concerning relationship between HI, TQR, and TL is presented in Table 3.

### 3.2. The Relationship of HI and TQR with the Physical, Physiological, and Technical Aspects

The research that has investigated the influence of well-being and recovery status on the physical, physiological, and technical aspects of soccer is very limited. Indeed, the increase in fatigue and insufficiency of recovery can cause neuromuscular disturbance and a decrease in technical efficiency [50]. In addition, it is suggested that the well-being and recovery state could be the main mediator variable between fitness and motor performance [51] and a psychometric imbalance alters the physiological and technical aspects specific to soccer [52]. Moreover, Selmi et al. [46] showed that participants with higher technical and physical levels were in good psychometric condition by indicating that the feeling of fatigue compromises technical activities such as passes and interceptions among soccer players. The authors also suggested that wellness indices and recovery status corresponding to physical performance declines lead to poor neuromuscular coordination, which has negatively influenced technical execution such as the accuracy of passes or other football qualities. In the same perspective, Selmi et al. [47] reported that the weak technical activity contributed to the signals of fatigue and muscle pain indicating that the feeling of fatigue has negatively affected nerve conduction and muscular coordination. In addition, Ferraz et al. [53] indicated that the feeling of fatigue and psychometric disruption may reduce performance and affect engine processing and the perceptual dispensation associated with the necessary execution skills in game situations.

These findings suggest that physiological, physical, and technical aspects can be influenced by TL, fatigue accumulation, the balance between training and recovery, and physical fitness [46]. Furthermore, studies have reported that psychometric disturbances affect sport performance.

In this perspective, the study by Selmi et al. [16] examined the relationships between TL-variables (i.e., TL, monotony, and strain), psychometric status (HI and TQR), physiological responses, and technical aspects determined in professional soccer players during the intense training cycle of early season preparation period (4 weeks). The major findings of this study were that physiological variables did not change after an intense training cycle and were not influenced by HI and TQR. Several relationships were found between some technical actions (successful passes, interception, and lost passes), TL and TQR fatigue and DOMS during the training period. These results suggested that TL, HI, and TQR represent useful strategies for coaches to control technical aspects in soccer players during the early season preparation period in soccer players [16].

Other studies [43,54] have placed interest on the relationship between the intensity of training exercises and the TQR. Brink et al. [43] used the TQR method to determine if recovery contributed to the prediction of physical performance. The results showed that the TQR did not contribute to the performance of a maximum aerobic power test [43]. Fanchini et al. [54] examined the effect of the intensity of the exercise during a training session on the assessment of the RPE and the TQR in young soccer players. Each session consisted of three blocks of 20 min of different intensities (low, moderate, and high), executed in a random order. The values of the perception of the effort and the TQR were not different between the conditions (*p* = 0.57, *p* = 0.55, and *p* = 0.96, respectively). In addition, Osiecki et al. [7] assessed the relationship between the state of the recovery and serum creatine kinase (CK) level in professional soccer players during an official competition. They showed a significant correlation between TQR and CK (r = −0.75; *p* < 0.05) indicating that TQR can be used in the assessment of the recovery state in professional athletes after an official match.

Charlot et al. [55] identified physiological and perceptive responses in high-level players during a large international football tournament. The results of this study have shown that the physiological parameters and level of fatigue, sleep quality, stress, and DOMS have been largely influenced by successive competitions. Changes of these indices are related to the increase in RPE and heart rate [55]. The study also showed significant correlations between HI measured before the match and the recovery state and heart rate measured after the match. In addition, fatigue appeared in players after the third game, although the quality of sleep, the level of fatigue, the DOMS and the stress were not changed significantly before and after the last match [55]. However, it was also suggested that the decrement of intensity is in relation with the bad ranking that is decided at the end of the third game and the lack of motivation of the players [55]. This could explain that psychological disturbance affects the technical, tactical, and physical performance of football players [56,57].

On the other hand, Nedelec et al. [58] studied the relationship between game actions performed during a soccer match and subjective markers in 10 professional soccer players during four competitions. In this study, the analyses of the time motion was made to determine the number of game actions performed by the players. The HI and the TQR were obtained before the match, and 24, 48, and 72 h after the match [58]. Significant correlations were found between the increase in DOMS and the number of sprints (5 m) performed during the match at 48 and 72 h [58]. These results suggested that HI and TQR can help coaches and physical coaches adjust the TL after a soccer match [58].

Another study [32] put the interest of the relationship between physical aspects (total high-intensity running distance, countermovement jumps height (CMJ), post-exercise heart rate recovery, heart rate variability, and HI) in elite soccer players during an in-season competitive phase (17 days). Results showed relationships between fatigue score, CMJ, and total high-intensity running distance, and the heart rate recovery was influenced by DOMS and sleep quality. The authors indicated that HI and heart rate variability were sensitive to daily fluctuations in total high intensity running distance for a sample of elite soccer players. They suggested that the use of these markers may be considered a particular promise as simple, noninvasive assessments of fatigue status in elite soccer players during an in-season competitive phase.

In addition, study of the relationship between TQR, HI, technical aspects, and physiological aspect during small-sided games (SSG) in professional soccer players [47] indicated that decreased self-reported recovery quality and increased pre-fatigue signs were not influenced by the physiological variables, but they were related to poor technical execution (i.e., successful passes, lost passes, tackles, and interceptions). This is indicating that poor recovery quality and higher HI can be associated with the performance of technical skills in soccer matches [47]. Therefore, these results indicate that the TQR scale and HI scale was sensitive to recovery state and could also be useful for predicting performance decline and optimizing training [47].

A summary of studies concerning relationships between HI, TQR, physiological, and technical aspects in soccer is presented in Table 4.

### 3.3. Influence of HI and TQR on the Feeling States

RPE can be used as an objective tool sensitive to the detection of the internal intensity of the effort and adapted to control the TL [9,11,15,16,20,25,47]. Thus, the use of the physical enjoyment questionnaire during the training is a useful strategy for technical staff to detect an emotional response and evaluate a positive feeling of training sessions.

Some studies have shown that HI and TQR do not influence RPE and physical enjoyment. In this context, Haddad et al. [20] indicated no relationship between well-being indices (i.e., sleep, stress, fatigue, and DOMS) recorded before the training sessions and the RPE during a submaximal effort. They suggested that the RPE reflects only the intensity of the training and not the individual signs of pre-fatigue. The authors indicated that the subjective assessment of sleep, stress, level of fatigue, and muscle pain is used to detect the current state of the athletes and express the negative sensation of adaptation to the training session [16,20].

In the same context, the study by Osiecki et al. [7] determined the relationship between the recovery state and the RPE in 10 players after a professional football match. The results showed no significant association between the TQR and the RPE. The authors indicated that the TQR could be used in the evaluation of professional soccer players to determine the recovery state after an official match [7].

A recent study reported that the TQR and the HI recorded before training session were not related to physiological variables and RPE during soccer specific training in professional players [47]. These results showed that RPE does not appear to be affected by variability of the HI and TQR during exercises training. Stress, fatigue, sleep, DOMS, and TQR are not contributing signals to altered RPE. This study offers support for the efficacy and utility of using simple HI and TQR measures as part of an overall program to monitor athlete readiness.

In the same context, Haddad et al. [20] studied the effects of the variation of HI on the RPE in 17 young soccer players during a training exercise. The results of this study showed no influence of HI in the RPE. They found that these indices are not the main contributors of the RPE in players during soccer training in the absence of an excessive TL [20].

The study of Selmi et al. [5] evaluated the effects of HI on physical enjoyment and RPE during SSG (four sessions SSG 4 vs. 4 are made at different days on a surface of 25 × 35 m) in the competitive period in professional soccer players. The results showed that physical enjoyment and RPE do not seem to be influenced by the variability of the quality of sleep, stress, fatigue, and DOMS during SSG [5]. These results suggested that the well-being indices measured before the training sessions are used to detect the current form of players, sensitive to the identification of stress after an intense training and adapted to monitoring short-term emotional changes [16,17,18,19,20].

A second study [46] examined the influence of the TQR on physical enjoyment during SSGs in young soccer players. The results of this study revealed that physical enjoyment does not seem to be influenced by the variability of the TQR [16]. This finding suggested that the TQR does not transmit signals to change the enjoyment of the players during SSG [46]. The authors also indicated the contribution to SSG is motivating and encouraging. They conclude that these forms of training aroused much more motivation among players than other exercises [1,47,59].

Moreover, Selmi et al. [1] indicated that physical enjoyment measured during specific soccer exercises in young soccer players was not affected by HI variability. This finding suggests that the variables measured by these scales do not contribute to physical enjoyment during training session. This study also revealed that physical enjoyment was not influenced by psychometric variables, suggesting that positive feeling is also unrelated to well-being indices.

A summary of studies concerning the influence HI and TQR on the feeling states in soccer is presented in Table 5.

### 3.4. Relationship between HI, TQR, and Ramadan

The relationships between the month of Ramadan, psychometric states and muscle injuries were also subject to some scientific studies. To this end, Chamari et al. [30] studied the effects of Ramadan’s monthly workout on injury rates and HI in professional players during two consecutive seasons. Each year, players were monitored for 3 months: 4 weeks before Ramadan, during the month of Ramadan (4 weeks), and 4 weeks after Ramadan. This study showed no significant differences between the three periods in training load variables (i.e., weekly TL, training strain, training monotony) sleep quality, stress, fatigue level, and DOMS. However, they showed significant differences in pre-fatigue indices between the two groups, indicating that fasting players had a lower HI level and stress during and after the month of Ramadan compared with nonfasting players [30]. They suggested that the perception of sleep was not affected by the month of Ramadan, thus indicating that this quality is characterized by the different phases at the beginning and at the end of the night [30]. However, no significant difference in injury rates was observed between fasting and nonfasting players.

In addition, Boukhris et al. [60], investigated the effects of Ramadan fasting on HI following a soccer match simulation in professional soccer players. In this study the players performed a physical test (modified Loughborough intermittent shuttle test) on two occasions: one week before Ramadan and during the fourth week of Ramadan (End-Ramadan). The subjective ratings quality of sleep, fatigue, muscle soreness and stress were assessed at baseline and 0, 24, 48, and 72 h following the physical test. The results indicated that stress increased only at 0 h on end-Ramadan, while fatigue level increased at 24 h before Ramadan and at 0, 24, and 48 h at end-Ramadan with DOMS increasing throughout the recovery period at both occasions, with a higher level at end-Ramadan.

Bouzid et al. [61] subjected eight elite soccer players (age: 21.0 ± 0.4 years) to a modified protocol of Loughborough intermittent shuttle test (LISTmod), the first week and during the fourth week of Ramadan, in order to evaluate the effects of Ramadan fasting on recovery after a simulated football match. After LISTmod, the performance of the squat jump, the countermovement jumps at maximum voluntary contraction, and the 20 m sprint decreased significantly in both measures, but these decreases were greater at the end of the fourth week of Ramadan. Additionally, creatine kinase, uric acid, and muscle pain levels increased significantly. The fatigue rating increased in both week 1 and week 4 of Ramadan, while the stress rating only increased after 4 weeks of Ramadan. Hence the perturbations in physical performance and subjective evaluation parameters were highest at the end of Ramadan. However, the results of this study showed that Ramadan fasting did not negatively impact recovery after simulating a football match in professional footballers [61].

A summary of studies concerning the influence of HI and TQR on Ramadan in soccer is presented in Table 6.

The main limitations of the present systematic review are not related to the low number of studies investigated on this subject, but to the fact that most of them had a relatively small number of participants. Additionally, it is essential to take into consideration that the studies analyzed were conducted in soccer players with different levels of physical activity, ages, and using different research protocols, which increases the heterogeneity between studies.

The future inclusion of other psychometric indices of fatigue, recovery, and sleep, and other sports disciplines, could enrich the vision of important variables that could influence sports performance.

In conclusion, lack of recovery, poor quality of sleep from the previous night, and increased levels of stress, fatigue, and DOMS negatively impact athletic performance [16,21]. Since HI and TQR recorded before each training or match are affected by the variability of the training load (TL), they can influence the physical and technical performance and affective aspects of footballers. Monitoring the psychometric state of recovery of the players before each training or match to detect the first signs of fatigue allows for better detection of the signs of individual fatigue, and also allows the coaches to adapt and readjust the TL. This avoids physical and technical gaps and improves the performance of the players. Successful training must avoid the combination of excessive overload and inadequate recovery; otherwise, athletes can experience short-term decline in performance without severe physical and psychological symptoms or other lasting negative symptoms [21,62,63,64]. In fact, when athletes do not respect sufficiently the balance between training and recovery, an overreaching syndrome or nonfunctional overtraining can occur, with maladjustment not only of the player (such as fatigue, decline in performance, and mood disorders), but also of various biological factors such as neurochemical and hormonal regulation mechanisms [63,64,65].

For example, exercise-induced muscle damage may occur in players [64], leading to the onset of an inflammatory response associated with decreased ability to generate muscle force, reduced range of motion, localized swelling, delayed-onset muscle soreness, and increase in muscle proteins in the blood (creatine kinase, lactate dehydrogenase, and myoglobin [64].

## 4. Conclusions

This review included the relationships between psychometric status, HI, and TQR recorded prior to each training session or match and the training load and technical and physical performance in the players.

Therefore, it offers support for the effectiveness and usefulness of using simple HI and TQR measurements as part of an overall program to monitor the athlete’s readiness. Our results also highlight how those subjective feelings of negative well-being (i.e., perceptions of sleep, stress, fatigue level, and DOMS) and recovery state have an unfavorable impact on player performance

## 5. Practical Applications

This review was conducted to support the interest in measuring HI and TQR before the daily first training session. HI and TQR are simple noninvasive, nonfatiguing, sensitive, and effective methods that are able to help coaches and physical coaches in monitoring psychometric status and TL of soccer players; both parameters may be useful in predicting players’ readiness during training. It follows that coaches and physical coaches should keep in mind that increased TL, poor recovery, and fatigue acclimation negatively affect the psychometric status of the players. The increase in the pre-fatigue signal (HI and TQR) is in fact effective for evaluating the performance of players and represents a useful method for coaches and physical trainers to control changes in technical, tactical, and physical performance during training sessions, training period, or match.

This method not only allows better identification of signs of individual fatigue, but also allows coaches and physical trainers to choose the best program, types, and qualities of exercises for the players, and to plan the workload precisely to achieve better performance.

## Figures and Tables

**Table 1 ijerph-19-09385-t001:** Hooper indices (Hooper, 1995 [21]).

Sleep	Stress
1- very, very good	1- very, very low
2- very good	2- very low
3- good	3- low
4- medium	4- medium
5- bad	5- high
6- very bad	6- very high
7- very, very bad	7- very, very high
**Fatigue**	**DOMS**
1- very, very low	1- very, very low
2- very low	2- very low
3- low	3- low
4- medium	4- medium
5- high	5- high
6- very high	6- very high
7- very, very high	7- very, very high

DOMS: delayed-onset muscle soreness.

**Table 2 ijerph-19-09385-t002:** Total quality of recovery (TQR).

Total Quality of Recovery (TQR)	
6	-
7	Very-very low recovery
8	-
9	Very-low recovery
10	-
11	11 Low Recovery
12	-
13	13 Reasonable recovery
14	-
15	15 Good recovery
16	-
17	17 Very good recovery
18	-
19	Very-very good recovery
20	-

**Table 3 ijerph-19-09385-t003:** Relationship between Hooper index (HI), total quality of recovery (TQR), and internal training load/training periods.

Study	Participant (Number, Sex, Level, Age)	Index Measure		Condition/Duration	Aim	Results	Findings
		HI	TQR				
Selmi et al. (2018b) [16]	16, male, professional (25 ± 0.8)	Y	Y	Mid-season competitive period (4 weeks)	To investigate the effect of training load of early season preparation period on psychometric status	TL was associated to HI scores (sleep, stress, fatigue, and DOMS).	Sleep, stress, fatigue, and DOMS represent a useful strategy for coaches to control TL in soccer players during early season preparation period.
Nobari et al. (2020) [10]	29, male, professional (15 ± 0.2)	Y		From the beginning of competitive period for eighteen weeks and two weeks half a season	To analyze the associations between training load metrics and weekly (w) reports of HI scores (sleep, stress, fatigue, and DOMS)	There is a correlation between TL variables (TL, monotony, and strain) and weekly well-being indicators between the 20 weeks.	HI scores moderate-large related to acute load, monotony, and strain; however, overall weekly HI was the best predictor of the acute load.
Perri et al. (2021) [45]	28, male, subelite (20.9 ± 2.4)	Y		Competitive season	To investigate the relationship between the daily training load and HI	A significant correlation was reported between daily TL and HI measured the day after; additionally, a similar weekly pattern seems to be repeating itself throughout the season in both TL and HI.	TL affects the HI in soccer players.
Clemente et al. (2017) [9]	35, male, professional (25.7 ± 5.0)	Y		Entire competitive period	To examine the relationship between TL and HI (sleep, stress, fatigue, and DOMS) across two different training microcycles (1 vs. 2 competitive games)	DOMS, fatigue and HI were higher in 2-game weeks compared with 1-game weeks. ITL was negatively correlated to DOMS, sleep, fatigue, stress, and HI in 2-game weeks. From 1-game microcycle only TL negatively correlated to stress.	As a result, care should be taken when planning the lead into and out of a 2-game fixture microcycle, highlighting key specific recovery strategies to dampen the increased stress effect.
Nobari et al. (2021) [11]	36, male, elite (15.5 ± 0.2)	Y		Entire season	To determine weekly (w) and daily variations of well-being ratings relative to HI (i.e., fatigue, stress, DOMS, and sleep quality) during a soccer season based on players’ positions	There were found: A significant increase in stress and sleep for all players’ positions from early- to end-season. A significant difference between well-being status 5 days before match day (MD) and 4 days before MD, compared to MD for all playing positions. The highest and lowest records occurred during end-season for fatigue (central midfielders, and for DOMS (strikers), early season (central defenders), and early season (wide defenders).	Coaches must use HI for monitoring their teams throughout the full season, to avoid overtraining and injuries.
Lathlean et al. (2019) [48]	562, male, elite (17.7 ± 0.3)	Y		One competitive season	To investigate associations between TL (training and competition) and HI (sleep, stress, fatigue, and DOMS)	Season overall wellness had a significant linear negative association with 1 week TL and an inverse U-curve relationship with session TL. HI scores were identified to have associations with TL.	TL is important in managing the HI of players. Quantifying TL and HI helps to optimize player management and has the potential to avoid injury.
Malone et al. (2018) [49]	48, male, professional (25.3 ± 3.1)	Y		Entire competitive season	To investigate the relationship between training and HI (sleep, stress, fatigue, and DOMS) in response to training and/or match load	Significant effects of HI scores on integrated and external TL measures.	Monitoring of player HI can offer coaches with information about the training program that can be usual from individual players during a training session.
Thorpe et al. (2015) [32]	10, male, elite, (19.1 ± 0.6)	Y		In-season competitive period	To determine the sensitivity HI measures to daily TL accumulated over the previous 2, 3, and 4 days (d) during a short in-season competitive period	Correlations between variability HI were negligible and not statistically significant for all accumulation TLs.	The sensitivity of HI variables to changes in TL is generally not improved when compared with TLs.
Pereira et al. (2022) [14]	18, male, professional (24.3 ± 4.8)	Y	Y	Entire season	To analyze TL and TQR and HI changes in professional soccer players after a 4 week pre-season	Higher TL values associated with reduced TQR and increased DOMS scores.	Strong, positive associations between TL and psychometric indices (TQR, DOMS)
Fessi et al. (2016) [25]	17, male, professional (23.7 ± 3.2)	Y		Competitive period	To explore changes in weekly TL, quality of sleep, quantity of stress, fatigue, DOMS, and affective valence between pre- and in-season periods of professional soccer players	Higher players’ TLs were recorded during pre-season when compared with in-season period. The ratings of sleep, stress, fatigue, and DOMS in pre-season were higher than those observed during in-season whereas the feeling score was lower.	Pre-season period of training induces significantly more strenuous and exhausting demands on professional soccer players compared with the in-season period at the elite level.
Moalla et al. (2016) [15]	14, male, professional (25.7 ± 2.6)	Y		At the beginning of the 2013–2014 season (16 week training period)	To investigate the relationship between daily TL and the HI (sleep, fatigue, stress, and DOMS)	Significant relationships between TL and perceived fatigue, muscle soreness, sleep, and stress.	HI is both a simple and useful tool for monitoring perceived well-being and psychometric players’ status of professional soccer players.
Selmi et al. (2021) [3]	15, male, professional (24 ± 1)	Y	Y	During the pre-season	To examine the perceived well-being, TQR, and psychological responses during an intensified training period (IT)	Significant relationships were found between TL and HI, TQR.	HI and TQR found to be sensitive measures and may provide coaches with information about wellness and psychological state of soccer players during IT.
Nobari et al. (2020) [10]	21, male, elite (under 16 years old)	Y		Competitive season	To analyze the associations between TL, monotony, strain, and HI	HI indicators were moderate-large related to TL, monotony, and strain.	HI was the best predictor of the acute load.
Selmi et al. (2020) [1]	15, male, professional (24 ± 1)	Y	Y	Intensified training periods (IT) (2 weeks)	To examine the relationship between TL, HI, and TQR during intensified training period (IT)	TL, monotony, and strain increased during IT HI (stress, sleep quality, fatigue level, and DOMS) increased and TQR decreased during IT. TL related to HI, TQR.	Higher TL affect negatively, perceived well-being, recovery state of soccer players during IT
Fernandes et al. (2021b) [26]	19, female, professional (24.1 ± 2.7)	Y		In-season period (10 weeks)	To describe the association between weekly variations of TL, monotony, strain, and weekly variations of HI (stress, fatigue, DOMS, and sleep)	Some associations between HI categories (sleep, stress, fatigue, DOMS) and TL variables.	Higher TL associated with higher HI.
Fernandes et al. (2021a) [44]	16, female, professional (24.0 ± 2.9)	Y		Competitive season	To compare session rated HI between training and match days (MD) from the same women’s Portuguese League team	DOMS revealed differences between MD-4 vs. MD-2; HI showed higher values on MD-5 vs. MD-4 vs. MD-2 vs. MD.	Results from HI showed that sleep, fatigue, stress, and DOMS were fairly well controlled by coaches and staff.

Y: yes, TL: training load, HI: Hooper index; TQR: total quality of recovery, RPE: rating perceived exertion, SSG: small-sided games, DOMS: delayed-onset muscle soreness, s-RPE: session-RPE.

**Table 4 ijerph-19-09385-t004:** The relationship between Hooper index (HI) and total quality of recovery (TQR) on physiological and technical aspects in soccer.

Study	Participants (Number, Sex, Level, Age)	Index Measure		Condition/ Duration	Aim	Results	Findings
		HI	TQR				
Selmi et al. (2018b) [16]	16, male, professional (25 ± 0.8)	Y	Y	Early season preparation period (4-weeks)	To investigate the influence of HI and TQR on physiological and technical aspects during an intense training cycle	Physiological variables did not change after IT and were not influenced by HI and TQR. HI and TQR were related to successful passes, interceptions, and lost passes measured after IT during soccer specific training test.	No relationship was recorded between psychometric state (HI and TQR) and physiological responses during soccer specific training and those technical aspects were affected by the TQR and the HI variability.
Buchheit et al. (2014) [17]	18, male, professional (21.9 ± 2.0)	Y		Pre-season training camp	To examine the usefulness of physiological and psychometric variables during high-intensity running performance	Changes in submaximal exercise HR (Hrex) and sleep, stress, fatigue level, DOMS, but not cortisol, were slightly to very largely correlated with changes in Yo-YoIR2 performance and HSR during the standardized training drills.	Sleep, stress, fatigue level, DOMS, and HRex, but not cortisol, are highly sensitive to subtle daily changes in TL and are well correlated with positive changes in high-intensity running performance
Selmi et al. (2021) [3]	15, male, professional (25 ± 1)	Y	Y	Intensified session (2 weeks)	To examine the relationship between HI (sleep, stress, fatigue, and DOMS), TQR, countermovement jump, and biochemical markers of fatigue in response to an intensified training period	HI was positively correlated with cortisol, T/C ratio, and creatine kinase, and negatively correlated with CMJ. Furthermore, TQR was negatively correlated with T/C ratio, creatine kinase, and C-reactive protein, and positively correlated with CMJ.	Neuromuscular fatigue, muscle damage, and change in the anabolic/catabolic state induced by the IT were related to well-being and TQR among professional soccer players.
Clemente et al. (2021) [4]	25, male, professional (28.1 ± 4.6)			Pre-season period	To analyze the blood measures changes and their relationships with HI changes after pre-season training.	Correlations were found between HI, all derived RPE measures, hematological variables, and biochemical measures.	The results indicated the significant relationships between blood and well-being measures; monitoring hematological and biochemical measures allow coaches to minimize injury risk, overreaching, and overtraining.
Mendes et al. (2022) [34]	35, male, professional (25.7 ± 5.0)	Y		Entire season	To determine the relationships between the HI and CK levels over the weekly microcycles of the season	HI and CK were significantly higher in weekly microcycles with one match than with two.	HI would be a very useful approach to monitor the effects of TL in elite professional soccer players.
Thorpe et al. (2015) [32]	10, male, elite players (19.1 ± 0.6)	Y		Competitive period (17 days)	To quantify the relationship between HI and total high intensity running distance (THIR), countermovement jump height (CMJ), and heart rate variability	Fluctuations in fatigue were significantly correlated with THIR distance. Correlations between variability in muscle soreness, sleep quality, heart rate variability, and THIR distance were negligible and not statistically significant.	Perceived ratings of fatigue were sensitive to daily fluctuations in THIR distance.
Saidi et al. (2022) [12]	14, male, elite soccer players	Y		Congested period of match play (12 weeks)	To analyze HI, biochemical markers and physical fitness in relation to changes in training and match exposure	A significant increase was found in stress, fatigue, DOMS, and HI during the congested period of match play (CP) compared with the regular period of match play (RP). In CP, significant relationships were found between C-reactive protein and creatine kinase with the HI, and the fatigue score. In addition, the fatigue score and DOMS correlated with Yo-Yo intermittent recovery test and best of repeated shuttle sprint ability test.	Elite soccer players’ well-being status reflects declines in physical fitness during intensive period of congested match play, while biochemical changes do not.
Osiecki et al. (2015) [7]	10, male, professional (26.6 ± 4.5)		Y	Competitive season	To indicate the relationship between TQR, RPE, and creatine kinase (CK) after an official professional soccer match	No significant associations were found between TQR and RPE; CK and RPE. However, we did find a statistically significant association between TQR and CK.	The findings indicate that TQR could be used in the evaluation of professional soccer players to determine recovery state after an official game.
Brink et al. (2010) [43]	18, male, elite (17 ± 0.5)		Y	Full competitive season	To investigate the relation between TL, TQR, and monthly field test performance	Session RPE and TQR scores did not contribute to the prediction of performance. The duration of training and game play in the week before field test performance is most strongly related to interval endurance capacity.	Coaches should focus on training duration to improve interval endurance capacity in elite soccer players.
Nedelec et al. (2014) [58]	10, male, professional (21.8 ± 3.2)	Y	Y	From mid- to end-season.	To examine the relationship between, HI, TQR, CMJ, isometric maximum voluntary contraction (MVC) of the hamstring muscles, peak speed (PS) and playing actions completed during the match (PACM)	Correlations between CMJ, MVC, PS, fatigue, muscle soreness, TQR, and PACM were assessed. Significant correlations were observed between the DOMS and the number of sprints <5 m performed during the match at 48 and 72 h.	Fatigue, DOMS, and TQR affect neuromuscular fatigue and physical aspects for up to 72 h.

Y: yes, TL: training load, HI: Hooper index; TQR: total quality of recovery, RPE: rating perceived exertion, SSG: small-sided games, DOMS: delayed-onset muscle soreness, s-RPE: session-RPE, GPS: global positioning system.

**Table 5 ijerph-19-09385-t005:** Influence of the Hooper index (HI) and total quality of recovery (TQR) on the feeling.

Study	Participant: (Number, Sex, Level, Age	Index Measure		Condition/Duration	Psychological Variable Measured	Aim	Results	Findings
		HI	TQR					
Selmi et al. (2018a) [5]	16, male, professional (25 ± 0.8)	Y	Y	Mid-season competitive period (4 weeks)	Physical enjoyment (PACES)	To assess the effects of the HI on physical enjoyment (PE) and RPE during soccer specific training sessions	PE was not related to HI variables (sleep, stress, level of fatigue, and DOMS) and RPE.	Rating of PE and RPE does not seem to be influenced by the variability of HI during SSG with young players
Selmi et al. (2018d) [47]	16, male, professional (16,5 ± 0,6)	Y		Last 3 weeks of the competitive season	Physical enjoyment (PE)	To examine the effects of the HI on physical enjoyment (PE) during SSG	The rating of PE does not seem to be influenced by the variability of the HI during SSG with young players. Stress, fatigue, sleep, and DOMS are not contributing signals to altered PE.	The PE induced by a training method might vary according to modality of exercise, outcomes, and desire of the players.
Selmi et al. (2018c) [46]	16, male, young soccer players (16.5 ± 0.6)		Y	Competitive season	Physical enjoyment (PE)	To investigate the effects of the TQR on physical enjoyment (PE) rating during SSG	No significant correlation found between TQR and PE.	The PE induced by a training method might vary according to types of exercise, motivation, and encouragement of the players. PE does not seem to be affected by the variability of TQR during SSG.
Selmi et al. (2020) [1]	15, male, professional (24 ± 1)	Y	Y	Intensified training periods (IT) (2 weeks)	Profile of mood states (POMS)	To examine the perceived well-being, recovery quality and psychological responses during intensified training period (IT)	Significant relationships were found between TL and HI, TQR and mood state.	HI?, TQR, and mood were found to be sensitive measures and may provide coaches with information about psychological state of soccer players during IT.
Fessi et al. (2016) [25]	17, male, professional (23.7 ± 3.2)	Y		Competitive period	Feeling scale	To examine the association between HI and affective valence during pre- and in-season	Affective valence associated with sleep, stress, fatigue, and DOMS during pre- and in-season.	PE seem to be affected by the variability of HI

Y: yes, TL: training load, HI: Hooper index; TQR: total quality of recovery, RPE: rating perceived exertion, SSG: small-sided games, DOMS: delayed-onset muscle soreness, s-RPE: session-RPE.

**Table 6 ijerph-19-09385-t006:** Effect of Ramadan in Hooper index (HI) and total quality of recovery (TQR).

Study	Participant: (Number, Sex, Level, Age)	Condition/ Duration		Aim	Results	Findings
		HI	TQR			
Chamari et al. 2012 [30]	42, male, professional	Y	Ramadan (during two consecutive seasons)	To determine the effects of Ramadan’s monthly workout on injury rates and HI in professional players	This study showed no significant differences between the three periods (4 weeks before Ramadan, during the month of Ramadan, and 4 weeks after Ramadan) were observed in sleep quality, stress, fatigue level, and DOMS	HI was not affected by the month of Ramadan.
Bouzid et al. 2019 [61]	8, male, elite soccer players (21.0 ± 0.4)	Y	Ramadan month	To examine the effects of Ramadan fasting on HI following soccer matches simulation	Stress increased only at 0 h on end-Ramadan, while fatigue level increased at 24 h at before-Ramadan and at 0, 24, and 48 h at end-Ramadan, and DOMS increased throughout the recovery period at both occasions, with a higher level at end-Ramadan.	Subjective ratings parameters were higher at the end of Ramadan in soccer players.

Y: yes, TL: training load, HI: Hooper index; TQR: total quality of recovery, RPE: rating perceived exertion, SSG: small-sided games, DOMS: delayed-onset muscle soreness, s-RPE: session-RPE.

## Data Availability

The data presented in this study are available on request from the corresponding author.

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
