# Peer review of "Monitoring Psychometric States of Recovery to Improve Performance in Soccer Players: A Brief Review"

_ijerph, 2022, doi:10.3390/ijerph19159385_

Round 1

Reviewer 1 Report

Enrich the results and above all with the conclusions of your literature review

L45 I suggest also highlighting the recent increase in the role assumed as follows: “Among other things, even in a return-to-play (RTP) scenario it has become fundamental for elite athletes after an acute trauma, given the economic and competitive interweaving associated with the downtime of professional players, also the psychological facet of RTP after an injury.” Ref: https://doi.org/10.3390/medicina57111208

L83 Pubmed, Scopus, WoS, and Google Scholar

L86 Present the strings better

Restructure if you talk about materials and methods, as an outcome the scales presented and then the tables with the articles included in a results paragraph.

There seems to be a lack of an assessment of the quality of the manuscripts ... At least one Joanna Briggs? To also give arguments to the clinical implications. Among other things, before exposing the latter, present a section on limitations

Author Response

Dear

Editor in Chief

I would like to thank you for the internal evaluation on our manuscript.

We have made the suggested changes to our manuscript.

Hoping that the revised manuscript will be considered suitable for publication, I look forward to your editorial reply.

The revised manuscript has been proofread by a native English speaker

Yours Sincerely

Reply to Referee 1

  • L45: According to the referee's suggestion, we have added:

"Among other things, even in a scenario of return to play (RTP) it has become fundamental for elite athletes after an acute trauma, given the economic and competitive interweaving associated with the downtime of professional players, also the psychological aspect of RTP after an injury ".

  • L83: We corrected: Pubmed, Scopus, WoS, and Google Scholar
  • L86: as also suggested by referee 2 we have added other keywords to the search
  • We have also restructured the materials and methods and results paragraphs. Thanks to the referee's suggestions, the paper is now clearer
  • We assessed the quality of the manuscripts as specified in materials and methods:

"We used indexing words and free search terms, which we clustered according to the PICo scheme (population, phenomenon of interest, context) [28] ".

  • we have also added two paragraphs arguing the limitations of the present research and the clinical implications.

Reviewer 2 Report

the article is a current and relevant synthesis for the approached topic, of interest in obtaining the sports performance in football (soccer).

some suggestions that may increase the value of the article:

1. adding some keywords, to make a reference to the evaluation, recovery

2. In the Introduction, there are some references to the current state of knowledge and to other sports, although reference is made to football (see bibliographic sources 22, 23, 24).

3. most descriptions of the analyzed factors that influence the performance do not take into account the variable: gender, level of performance. perhaps a redistribution of information presentation would be useful.

4. in Tables 3-6, for an easier connection between the authors and the corresponding number in the References, add this corresponding number (for example, Selmi et al., 2018d [48]).

5. in each of the tables 3,4,6 there are references to sources that do not appear in the bibliography. for example, in table 3, to be checked for verification: Govus et. al., 2017, Gjaka et al., 2016, Douchet et al., 2021, Novack et al., 2021, Teixeir et al., 2021, Chavas et al., 2018). in table 4: Selmi .., 2019, Rey .., 2019, Zurutuza .., 2017, Wellman .. 2017, Tomazoli..2020, Kieran..2019. in table 6, Bouzid .. 2019, Chamari..2012

6. In the same tables, some years of the indicated sources do not correspond to what is written in the References. for example, table 6, Baklouti ... is written 2016 in the table, in References is 2017. in table 4, Buchheit et all. is 2013 and in references 2014.

7. to introduce a series of study limits

Author Response

Dear

Editor in Chief

I would like to thank you for the internal evaluation on our manuscript.

We have made the suggested changes to our manuscript.

Hoping that the revised manuscript will be considered suitable for publication, I look forward to your editorial reply.

The revised manuscript has been proofread by a native English speaker

Yours Sincerely

Reply to Referee 2

  • As suggested by the referee, we have added some keywords
  • In the Introduction, we have replaced some references with more appropriate ones
  • We agree with the referee, in fact this is one of the limitations of our study.

However, in this version we have added a paragraph related to the relationship between HI, TQR and internal training load, in female players

On the other hand, it is difficult to distinguish the information based on the level of performance, as the latter is very often not reported, however the data reported were all related to professional footballers.

4, 5 and 6) In tables 3 – 6, we deleted authors who are not cited in the text and bibliography, the years of publication were corrected, and we also added the corresponding numbers

Reviewer 3 Report

In tables 3 and 4 are cited authors who are not found in the bibliography .

The phrase from 360- 363 has nothing to do with the research, I recommend removing it.

Author Response

Dear

Editor in Chief

I would like to thank you for the internal evaluation on our manuscript.

We have made the suggested changes to our manuscript.

Hoping that the revised manuscript will be considered suitable for publication, I look forward to your editorial reply.

The revised manuscript has been proofread by a native English speaker

Yours Sincerely

Reply to Referee 3

  • We deleted authors who are not cited in the bibliography, from tables 3 and 4
  • We removed the phrase from 360- 363.

Round 2

Reviewer 1 Report

I can thank the authors for their review efforts